# Small Nucleolar Derived RNAs as Regulators of Human Cancer

**DOI:** 10.3390/biomedicines10081819

**Published:** 2022-07-28

**Authors:** Alexander Bishop Coley, Jeffrey David DeMeis, Neil Yash Chaudhary, Glen Mark Borchert

**Affiliations:** 1Department of Pharmacology, College of Medicine, University of South Alabama, Mobile, AL 36688, USA; abc1323@jagmail.southalabama.edu (A.B.C.); jdd1527@jagmail.southalabama.edu (J.D.D.); nyc1921@jagmail.southalabama.edu (N.Y.C.); 2School of Computing, University of South Alabama, Mobile, AL 36688, USA

**Keywords:** cancer, gene regulation, small nucleolar RNA (snoRNA), small nucleolar derived RNA (sdRNA), microRNA (miRNA), RNA, snoRNA, sdRNA, miRNA, genetics

## Abstract

In the past decade, RNA fragments derived from full-length small nucleolar RNAs (snoRNAs) have been shown to be specifically excised and functional. These sno-derived RNAs (sdRNAs) have been implicated as gene regulators in a multitude of cancers, controlling a variety of genes post-transcriptionally via association with the RNA-induced silencing complex (RISC). In this review, we have summarized the literature connecting sdRNAs to cancer gene regulation. SdRNAs possess miRNA-like functions and are able to fill the role of tumor-suppressing or tumor-promoting RNAs in a tissue context-dependent manner. Indeed, there are many miRNAs that are actually derived from snoRNA transcripts, meaning that they are truly sdRNAs and as such are included in this review. As sdRNAs are frequently discarded from ncRNA analyses, we emphasize that sdRNAs are functionally relevant gene regulators and likely represent an overlooked subclass of miRNAs. Based on the evidence provided by the papers reviewed here, we propose that sdRNAs deserve more extensive study to better understand their underlying biology and to identify previously overlooked biomarkers and therapeutic targets for a multitude of human cancers.

## 1. Introduction

Until the first association of microRNA (miRNA) dysregulation and cancer in 2002, noncoding RNAs (ncRNA) in cancer were largely disregarded by many as “junk RNA” or transcriptional noise [1]. miRNAs are ~22 nucleotide (nt) long small RNAs which form part of the RNA Induced Silencing Complex (RISC) and suppress messenger RNA expression through miRNA-mRNA sequence complementarity. The pathway in which miRNA facilitates post-transcriptional gene suppression of target mRNA is formally known as RNA interference (RNAi). Elucidation of this RNAi pathway afforded the scientific community with a broader understanding of the existence of short ncRNA species and their relevance with respect to regulation of protein coding gene expression, including tumor suppressor genes in the context of human cancer. While a plethora of studies confirmed roles for dysregulated miRNAs in cancer, others wondered if additional short ncRNA species with distinct functions could similarly arise from full-length ncRNA transcripts. In 2008, Kawaji et al. performed an unbiased investigation of small RNA sequencing data obtained from HepG2 cells. In this study, miRNA-like fragments were aligned to the human genome [2] which resulted in the first ever characterization of noncoding-derived RNA (ndRNA) excised from other full length ncRNAs including transfer RNA, ribosomal RNA, and small nucleolar RNA (snoRNA). Similar to miRNAs, many of the observed ndRNA species were initially disregarded as degradation products or transcriptional noise and were assumed to lack functional purpose. However, many studies have since demonstrated that ndRNAs can be specifically excised from longer ncRNA species and can exert microRNA-like regulatory function on target genes. This distinguishes ndRNAs from the canonical roles of the longer transcripts from which they were derived. That said, annotation of these novel ndRNAs supports the belief that the human ncRNA regulatory network is vastly larger than currently appreciated. Moreover, expansion of the repertoire of tractable ncRNA targets for biomarker detection and drug discovery to include the emerging classes of ndRNAs is imperative. Clinical recognition of ndRNAs which possess comparable roles in gene regulation and disease pathogenesis to miRNAs will expand our collection of diagnostic markers. This will hopefully contribute to earlier disease detection and thereby confer improved clinical outcomes. In this review we describe the functional role of ndRNAs which arise from snoRNA transcripts, henceforth referred to as sno-derived RNAs (sdRNAs), and how dysregulation of these RNAs contributes to oncogenesis and cancer progression. To date, sdRNAs have been identified to function similar to miRNA and PIWI-associated RNA (piRNA). Given the profound role of miRNA and piRNA with respect to post-transcriptional gene regulation and the extensive literature characterizing their influence in cancer, investigation of sdRNAs and their role in cancer is of utmost importance.

## 2. snoRNA Structure and Function

One class of ncRNA found to give rise to functional ndRNAs is the snoRNA. SnoRNAs constitute a class of ncRNA localized to the nucleolus that can be further divided into two main subgroups: C/D box and H/ACA box snoRNA [3]. C/D box snoRNA are structurally distinguished by the presence of a 5′ box C motif (RUGAUGA, R = purine) and its C’ box partner, as well as a 3′ box D motif (CUGA) with its D’ box partner. Self-complementarity drives the formation of the C/D box snoRNA stem structure that in turn binds accessory proteins NOP56, NOP58, 15.5 K, and fibrillarin to form the functional sno-ribonucleoprotein (snoRNP) (Figure 1 and Figure 2). Situated in the nucleolus, antisense domains located within the C/D box snoRNA serve as guides for the snoRNP to bind target ribosomal RNAs (rRNAs). The methyltransferase fibrillarin is then responsible for carrying out 2′-O-ribose-methylation of the rRNA [4,5,6]. H/ACA box snoRNA contain a 3′ ACA box (ACANNN, N = any nt) and two hairpins linked by an H box (ANANNA, N = any nt). The H/ACA box snoRNP complex is formed by association with NHP2, NOP10, GAR1, and dyskerin proteins (Figure 1 and Figure 3). The H/ACA snoRNA hairpins guide the snoRNP to target rRNAs where dyskerin carries out pseudouridylation [7,8,9]. Erroneous expression and/or activity of either of these classes of snoRNA have been extensively linked to tumorigenesis. However, reports have implied that several proteins comprising the snoRNP are required for the biogenesis of distinct species of sdRNA [10].

## 3. sdRNA Biogenesis

### 3.1. miRNA-Like Biogenesis of sdRNAs

Despite requiring the existence of a parental snoRNA for sdRNA biogenesis, the transcriptional abundance of any given sdRNA and its corresponding parent snoRNA are largely unrelated [52]. SnoRNAs are approximately 100–300 nt in length and primarily arise from intronic transcripts produced by splicing mechanisms [11,12,13,53]. However, there is evidence supporting that correct expression and cellular localization of H/ACA snoRNA are mediated by RNA polymerase II dependent production of the snoRNA precursor, as is the case for snoRNA U64 [14]. After generation of the parent snoRNA, processing and excision of the mature sdRNA is thought to follow the classic microRNA biogenesis pathway. Consistent with this, there is evidence suggesting that H/ACA box and C/D box snoRNA are the evolutionary ancestors of a subset of miRNA precursors [19,40]. Indeed, the mature sdRNA and mature miRNA are nearly indistinguishable in structure. The exception to this is that sdRNA length varies slightly based on whether the parental snoRNA belongs to the C/D box family (~27 nt) or to the H/ACA box family (17–19 nt) while miRNAs are defined as 21–22 nt in length [2,19,40,54]. The key difference lies in biogenesis, as sdRNAs (1) arise from snoRNA transcripts and (2) can arise in a distinct manner from miRNAs (Figure 1). However, the full-length snoRNAs are believed to be processed into sdRNAs along the canonical miRNA processing pathway. In this biogenesis pathway, full length snoRNAs are processed into shorter transcripts by the microprocessor complex composed of Drosha and DGCR-8, similar to the conversion of primary microRNAs into precursor microRNAs [54,55]. Following this, the snoRNA fragment is exported from the nucleus into the cytoplasm via Exportin-5 in a RAN-GTP dependent manner. Once in the cytoplasm, the snoRNA fragment is processed further by the enzyme Dicer, yielding the mature sdRNA which can then associate with Argonaute-2 to form the RNA Induced Silencing Complex (RISC) [15,54]. At this point, the mature sdRNA functions as a molecular guide, directing RISC to a target mRNA. Upon perfect complementary binding of the sdRNA to the target mRNA, RISC then cleaves the mRNA resulting in its degradation. Alternatively, an imperfect binding of the sdRNA to the target results in a translational blockade wherein the presence of RISC impedes translation. That said, some studies on sdRNA processing may hint to the existence of disparate mechanisms for the biogenesis of miRNA-like sdRNAs which proceed independently of the microprocessor complex and DICER. Indeed, recent reports have indicated that some sdRNA species can arise from a mechanism independent of the microprocessor complex [10]. In lieu of the microprocessor complex, biogenesis of this particular sdRNA, derived from SNORD44, requires the presence of NOP58 and fibrillarin, two proteins necessary for the formation of snoRNP. Others have reported DICER independent production of miRNA-like sdRNAs [54]. Nonetheless, the remaining processing events and functional implementation are identical to the aforementioned biogenesis pathway for miRNA-like sdRNAs.

### 3.2. Non-miRNA-Like Biogenesis of sdRNAs

In contrast to miRNA-like biogenesis of sdRNAs, other reports have suggested the existence of sdRNA biogenesis pathways which deviate from the aforementioned pathway. For instance, the production of piRNA-like sdRNAs follows a distinct biogenesis pathway to which snoRNA transcripts are exported to Yb bodies where the 3′ ends undergo enzymatic cleavage from Zucchini (ZUC), followed by attachment of the PIWI domain to the 5′ end. The intermediate piRNA-like sdRNA is then further processed by Papi-dependent Trimmer and is trafficked to the cytoplasm, and then can re-enter the nucleus to suppress transcription (Figure 1) [16,17]. That said, there may be additional alternative pathways utilized in the biogenesis of sdRNAs which have not been realized.

## 4. sdRNAs in Cancer

Considering the expanding role of sdRNAs in cancer and the growing scientific interest surrounding them, we have provided a review of the existing literature highlighting specific sdRNAs involved in malignant pathology. The sdRNAs presented here represent the best studied sdRNAs to date. For convenience, Table 1 compiles these into a readily available reference with key information about each sdRNA.

Previous reports [19,40], our own literature analysis, as well as a cursory in-house alignment of miRNAs to full-length snoRNAs collectively show that many ncRNA species currently defined as miRNAs are in fact sdRNAs. Since sdRNAs have been rapidly garnering more scientific interest, it is important to retrospectively re-categorize these miRNAs as the sdRNAs that they actually constitute. For this review, we have included several such sdRNAs that have an established role in cancer, regardless of whether the author at the time referred to the ncRNA as a miRNA or sdRNA. Previous misidentification is reported in Table 1, and each section figure shows the alignments of each of these “miRNAs” to full length snoRNAs in the human genome when relevant. For clarity, we have attached the prefix “sd/” to the beginning of the miRNA’s name to indicate that the miRNA is snoRNA-derived.

### 4.1. sd/miR-664

In 2019 Lv et al. published a paper focused on elucidating the tumor-suppressive role of sd/miR-664 in cervical cancer. Comparing sd/miR-664 expression in cervical cancer patient samples and control tissue revealed that the sdRNA is expressed at lower levels in cancer. Using the Si-Ha cell line as a model of cervical cancer, sd/miR-664 expression was found to enhance apoptosis and reduce cell viability and migration. In a Si-Ha mouse xenograft model, treatment with an sd/miR-664 mimic drastically reduced tumor volume and weight while enhancing tumor apoptosis. The binding target was predicted to be c-Kit using online target prediction software, and this was confirmed via the Renilla luciferase assay. Sd/miR-664 expression was negatively correlated with c-Kit expression, which further solidified the proto-oncogene c-Kit as a target of this tumor-suppressive sdRNA in cervical cancer [57].

To assess the role of sd/miR-664 in cutaneous squamous cell carcinoma (cSCC), Li et al. quantified the sdRNA’s expression by in situ hybridization in patient samples as well as cSCC cell lines [58]. Both analyses revealed an increase in sd/miR-664 in cSCC compared to control. Phenotypic assays showed increased cSCC viability, colony formation, invasion, and migration when sd/miR-664 levels were elevated in cell lines. Additionally, a murine cSCC xenograft model treated with sd/miR-664 mimic exhibited increased tumor volume compared to control. The 3′-UTR of IRF2 was predicted and then confirmed to be a binding target of sd/miR-664. Taken together, sd/miR-664 functions as an onco-miR in cSCC by downregulating IRF2 expression. While the role of IRF2 has not been established in cSCC, IRF2 has been identified as a tumor suppressor in lung cancer and gastric cancer and, paradoxically, as an oncogene in testicular embryonal carcinoma [78,79,80]. This adds a layer of complexity to sd/miR-664, showing that its overall impact on tumor progression can be tissue context-dependent.

In 2020 Li et al. measured sd/miR-664-5p expression in HCC patient cancer and normal samples as well as the HCC cell lines HepG2 and SUN-475. They found that sd/miR-664-5p is significantly downregulated in HCC patient samples as well as in the HCC cell lines compared to normal control. Sd/miR-664-5p was discovered to function as a tumor-suppressing RNA, reducing cell viability, invasion, migration, and enhancing apoptosis as measured by phenotypic assays in HepG2 and SUN-475 cells. Target engagement with the AKT2 transcript 3′-UTR was confirmed and expression of sd/miR-664-5p was negatively correlated with AKT2 expression. AKT2 (PKB) is an oncogene and a critical component of the PI3K/AKT growth-promoting pathway. Taken together, this study identifies sd/miR-664-5p as a tumor-suppressing RNA that exerts its effect by transcriptionally regulating AKT2 [56].

### 4.2. Sd/miR-1291

Building on their 2016 publication where they found sd/miR-1291 to be downregulated in pancreatic cancer patient tissues, Tu et al. published an article in 2020 to further investigate this mechanism of action in pancreatic cancer cells [59,60]. They found that increasing sd/miR-1291 reduced ASS1 levels in the ASS1-abundant L3.3 pancreatic cancer cell line and sensitized them to arginine deprivation in vitro. In addition, the glucose transporter GLUT1 mRNA has previously been shown to be a direct target of sd/miR-1291 in renal cell carcinoma (RCC), and Tu et al. confirmed this to be the case in pancreatic cancer cells as well with a resulting decrease in glycolytic capacity [61]. Further, sd/miR-1291 treatment sensitized pancreatic cancer cell lines to cisplatin. Taken together, sd/miR-1291 has a clear role as a modulator of cell metabolism to bring about tumor suppression in pancreatic cancer.

In 2019 Cai et al. investigated the role of sd/miR-1291 in prostate cancer. They found that sd/miR-1291 is significantly downregulated in both prostate cancer patient samples as well as prostate cancer cell lines compared to normal prostate [62]. Overexpression of sd/miR-1291 in prostate cancer cells lines inhibited cell proliferation and caused cell cycle arrest at G_0_/G_1_, and significantly reduced tumor weight and volume in a murine model of prostate cancer. In silico target prediction identified the 3′-UTR of MED1 as a sd/miR-1291 target, and MED1 protein levels were shown to be inversely correlated with sd/miR-1291 expression both in vitro and in vivo. MED1 has previously been shown to be a marker of poor prognosis for prostate cancer [81]. This study therefore places sd/miR-1291 as a tumor-suppressing sdRNA that regulates the MED1 oncogene to restrict prostate cancer progression.

In 2020 Escuin and colleagues investigated whether miRNAs can be used as biomarkers to identify sentinel lymph node (SLN) metastasis. Sixty breast cancer patients matched primary and SLN metastasis samples were sequenced. Differential expression analysis revealed that miR-1291 expression is significantly increased in the SLN metastasis. By separating the SLN metastases by molecular subtype, the authors found that sd/miR-1291 is upregulated in HR+ breast cancer compared to the HER2+ subtype. A bioinformatic pathway analysis predicted that sd/miR-1291 regulates WNT signaling, planar cell polarity/convergent extension (PCP/CE) pathway, β-catenin independent WNT signaling, diseases of signal transduction, and signaling by receptor tyrosine kinases (RTKs), among others. While these results strongly suggest that sd/miR-1291 is a tumor-suppressing RNA, patient survival analysis revealed that expression of this sdRNA alone could not significantly predict patient survival [63].

### 4.3. Sd/miR-3651

A 2020 publication aimed at identifying novel contributors to colorectal cancer (CRC) found that miR-3651, which arises from SNORA84, was overexpressed in 34/40 CRC patient tumors [65]. Model cell lines of CRC were observed to have ~3-fold overexpression of sd/miR-3651. Phenotypic assays revealed that sd/miR-3651 expression was positively correlated with CRC cell proliferation, and that reduction of the sdRNA enhanced apoptosis via deactivation of PI3K/AKT and MAPK/ERK signaling. Target prediction identified the 3′-UTR of tumor suppressor TBX1 as a sd/miR-3651 target, which was confirmed via the luciferase assay. Since TBX1 is a transcription factor that has an established role as a regulator of PI3K/AKT and MAPK/ERK pathways, these results show a coherent mechanism of action by which sd/miR-3651 inactivates TBX1 and thereby activates pro-growth mitogenic pathways to function as an oncogenic sdRNA.

Wang et al. assessed the expression of sd/miR-3651 in esophageal cancer patient samples and found that sd/miR-3651 expression is significantly downregulated in esophageal cancer [66]. Kaplan–Meier analysis of a separate esophageal cancer patient cohort (*n* = 108) revealed that low sd/miR-3651 expression was a marker of poor overall survival and poor disease-free progression. In contrast with studies implicating sd/miR-3651 as an oncogenic sdRNA, Wang et al. showed that sd/miR-3651’s role in esophageal cancer is tumor suppressive which indicates that the effect of this sdRNA is tissue context-dependent.

### 4.4. sd/miR-768

In 2012, Su et al. published a study in which they interrogated gastric cancer patient samples for miRNA expression. They found that sd/miR-768-5p, which arises from SNORD71 (Figure 4), was significantly downregulated in cancer samples [68]. A 2013 study by Blenkiron et al. identified sd/miR-768-5p as a YB-1 binding partner via CO-IP in MCF7 cells [67]. The authors proposed potential explanations for the YB-1/sdRNA interaction including YB-1 functioning as a ncRNA-sponge to prevent ncRNA-mediated transcriptional repression as well as YB-1 potentially playing a role in ncRNA biogenesis.

In that same year, Subramani et al. found that sd/miRNA-768-5p and its alternatively processed variant sd/miRNA-768-3p were both greatly downregulated in lung cancer and breast cancer cell lines in vitro [69]. Patient brain metastases originating from lung, breast, ovary, melanoma, liver, parotid gland, thyroid gland, and large cell tumors were also found to have downregulated sd/miRNA-763-3p. Overexpression of either sd/miRNA-768-5p or sd/miRNA-768-3p increased lung and breast cancer enhanced cell viability and chemoresistance. KRAS, one of the most commonly mutated oncogenes in human cancer, was confirmed to be a sd/miRNA-768-3p target.

### 4.5. sd/miR-1248

There is a clear need for additional reliable biomarkers to aid care providers in their decision on which treatment to apply to prostate cancer patients following radical prostatectomy. Aimed at meeting this need, Pudova et al. published a paper in 2020 focused on identifying differentially expressed miRNAs between prostate cancer patient tumors with or without lymphatic dissemination [64]. Among the nine miRNAs found to be significantly upregulated in prostate cancer with lymphatic dissemination, sd/miR-1248 was identified which arises completely from SNORA81.

### 4.6. hsa-sno-HBII-296B and hsa-sno-HBII-85-29

In 2015, Müller et al. performed small RNA sequencing on six PDAC tissue samples and five normal pancreas tissue samples to assess the expression of ncRNAs not typically found via microarray analysis [70]. They considered 45 noncoding RNAs identified as significantly down-regulated in PDAC, of which there were fourteen sdRNAs and a single sno-derived piRNA. The most downregulated sdRNA in PDAC, as determined by log_2_ fold change, was hsa-sno-HBII-85-29. Four sdRNAs of a total 78 ncRNAs were significantly upregulated in PDAC. The most upregulated sdRNA in PDAC, as determined by log_2_ fold change, was hsa-sno-HBII-296B. Subsequent analyses were focused on a miRNA identified in the study, however this work constitutes the only publication that characterizes differentially expressed sdRNAs focused specifically in PDAC. PDAC is a particularly difficult cancer to detect and treat, with a reported 5-year survival rate of just 11% [82]. Müller et al.’s work lays the foundation for many future studies aimed at further characterizing sdRNAs in PDAC which can potentially provide tractable biomarkers/therapeutic targets to reduce the lethality of this especially morbid cancer type.

### 4.7. Sno-miR-28

By measuring snoRNA expression following induced P53 activation, Yu et al. found significant downregulation of all snoRNAs associated with the SNGH1 snoRNA gene [71]. One sdRNA arising from SNGH1-associated sno28, sno-miR-28, was identified in complex with AGO via HITS-CLIP data and was found to be abundantly expressed in patient breast cancer tissue. Elevating TP53 expression in vitro consequently reduced sno-mir-28, and in silico and in vitro target engagement experiments revealed the P53-stabilizing protein TAF9B as a likely sno-mir28 target. Taken together, the authors proposed that sno-mir28 directly regulates TAF9B to bring about indirect repression of P53, forming a loop as P53 overexpression decreases sno-mir28 levels. Taqman expression analysis of matched breast cancer patient tumor and normal tissues revealed that SNHG1, SNORD28 and sno-miR-28 were all significantly upregulated in tumors. Furthermore, using the MCF10A cell line as a model of undifferentiated breast epithelium, the authors found that sno-miR-28 overexpression enhanced breast cancer cells’ proliferative capacity and colony formation. In sum, this study defined a role for sno-miR-28 as an oncogenic sdRNA heavily involved in suppressing the P53 pathway.

### 4.8. sdRNA-93

In 2017 our lab published a study in which we investigated the role of sdRNAs in the aggressive breast cancer phenotype [72]. RNAseq analysis identified 13 total full length snoRNAs differentially expressed (>7.5×) in MDA-MB-231 compared to MCF7, 10 of which consistently give rise to sdRNAs that associate with AGO in publicly available HITS-CLIP data. We elected to focus on the sdRNA arising from sno93 (sdRNA-93) (Figure 5) due to it displaying the highest differential expression in MDA-MB-231 (≥75×) and previous publications implicating sdRNA-93 with miRNA-like silencing capabilities in luciferase assays [83]. SdRNA-93 silencing in MDA-MB-231 reduced cell invasion by >90% while sdRNA-93 overexpression conversely enhanced cell invasion by >100% at the same time point. By contrast, sdRNA-93 silencing in MCF7 had no significant effect on invasion while overexpression resulted in a substantial ~80% increase in cell invasion. We then employed multiple in silico miRNA target-prediction algorithms which predicted with consensus that Pipox, a gene involved in sarcosine metabolism which has been implicated in breast cancer progression, is a likely candidate for regulation by sdRNA-93. We then confirmed sdRNA-93/Pipox 3′-UTR target engagement in vitro using the Renilla luciferase assay. Taken together, these results identified sdRNA-93 as a strong regulator of breast cancer invasion, particularly in the more aggressive MDA-MB-231 cell line, with Pipox as a verified cellular target. Moving forward, sdRNA-93 could serve as a potential target for breast cancer therapeutics and biomarker studies.

### 4.9. sdRNA-D19b and sdRNA-A24

A study published in 2022 by our lab identified 38 specifically-excised and differentially-expressed sdRNAs in prostate cancer [73]. We began by querying PCa patient TCGA datasets alongside TCGA-normal prostate using an in-house web-based search algorithm SURFR (Short Uncharacterized RNA Fragment Recognition). SURFR aligns next generation sequencing (NGS) datasets to a frequently updated database of all human ncRNAs, performs a wavelet analysis to specifically determine the location and expression of ncRNA-derived fragments (ndRNAs) and then conducts an expression analysis to identify significantly differentially expressed ndRNAs. Two sdRNAs, sdRNA-D19b and sdRNA-A24 (Figure 3), were among the most overexpressed in PCa patient tumors and were identified as AGO-associated in publicly available datasets so they were selected for further scrutiny. In vitro phenotypic assays in PC3 cells, a model cell line for castration-resistant prostate cancer (CRPC), revealed that both sdRNAs markedly increased PC3 cell proliferation and that sdRNA-D19b, in particular, greatly enhanced cell migration. When the sdRNAs were overexpressed alongside treatment with chemotherapeutic agents, both sdRNAs provided drug-specific resistances with sdRNA-D19b levels correlating with paclitaxel resistance and sdRNA-A24 conferring dasatinib resistance. Multiple in silico target-prediction algorithms provided a consensus prediction of the CD44 and CDK12 3′ UTRs as targets for sdRNA-D19b and sdRNA-A24 respectively, which we then confirmed in vitro via the Renilla luciferase assay. Taken together, this work outlines a biologically coherent mechanism by which sdRNAs downregulate tumor suppressors in CRPC to enhance proliferative/metastatic capabilities and to encourage chemotherapeutic resistance.

### 4.10. Sd/miR-140

Based on prior investigations linking high expression of the long-noncoding RNA MALAT1 with poor prostate cancer patient prognosis, Hao et al.’s 2020 paper focused on elucidating the mechanism of action by which MALAT1 brings about this effect [41]. In silico target prediction and in vitro confirmation via the Renilla luciferase assay identified miR-140 as a target of MALAT1, indicating that MALAT1 functions as a miRNA sponge to reduce sd/miR-140 bioavailability. This was further supported by RIP-seq analysis where direct engagement of MALAT1 and sd/miR-140 was confirmed. Target prediction indicated that sd/miR-140’s target is the 3′UTR of BIRC, and this was confirmed via the Renilla luciferase assay. Knockdown of MALAT1 inhibited the growth of prostate cancer both in vitro and in vivo, an effect that the authors suggest is brought about through sd/miR-140’s release from repression by MALAT1. This outlines a pathway where sd/miR-140 functions as a tumor-suppressing RNA and is tightly regulated by the lncRNA MALAT1.

### 4.11. Sd/miR-151

A study in 2020 by Chen et al. found that sd/miR-151 is downregulated in human prostate cancer cell lines [44]. In prostate cancer cells, Chen et al. demonstrated that sd/miR-151 overexpression inhibited cell proliferation, migration, and invasion; enhanced apoptosis; and sensitized cells to treatment with 5-FU, an antimetabolite chemotherapy. While no target prediction was performed, the authors did find that overexpression of sd/miR-151 reduces phosphorylation of PI3K/AKT. Even without a precisely mapped mechanism of action, this publication links sd/miR-151 with prostate cancer as a tumor-suppressive sdRNA.

### 4.12. Sd/miR-215

A 2015 study by Ge et al. found that sd/miR-215 expression was reduced in both epithelial ovarian cancer cell lines and patient tissue [42]. Sd/miR-215 decreased cell proliferation, enhanced apoptosis, and enhanced sensitivity to paclitaxel treatment. In addition, increased sd/miR-205 resulted in a decrease of cellular X-linked inhibitor of apoptosis (XIAP) mRNA albeit without confirmation of target engagement. Vychytilova-Faltejskova et al. found in 2017 that sd/miR-215-5p is significantly reduced in CRC patient samples and that low sd/miR-215-5p expression is associated with late clinical stages of CRC as well as poor overall survival for CRC patients [43]. Overexpression of sd/miR-215-5p reduced cell proliferation, viability, colony formation, invasion, migration, in vivo tumor volume, and weight while enhancing apoptosis. Epiregulin (EREG) and HOXB9 mRNA were confirmed to be two targets of sd/miR-215-5p. Both genes are involved in epithelial growth factor receptor (EGFR) signaling, a critical pathway exploited by CRC to promote tumor growth.

In 2020 Chen et al. found that sd/miR-215-5p is downregulated in prostate cancer patient samples and that lower sd/miR-215-5p expression predicted worse overall survival [45]. Sd/miR-215-5p expression was found to be inversely correlated with cell viability, migration, and invasion. Target prediction algorithms identified the PGK1 mRNA as a likely target, and in vitro protein quantification revealed an inverse relationship between sd/miR-215-5p expression and PGK1. A 2020 study by Jiang et al. found that the lncRNA lnc-REG3G-3-1 regulates sd/miR-215-3p availability by adsorption in lung adenocarcinoma (LAD) [46]. Overexpression of sd/miR-215-3p in LAD cells was found to reduce cell viability, invasion, and migration. Both Leptin and SLC2A5 mRNA were confirmed as targets for sd/miR-215-3p.

### 4.13. Sd/miR-605

With their 2011 publication, Xiao et al. found that sd/mir-605 overexpression significantly reduced cell survival and enhanced apoptosis in breast and colon cancer cell lines [47]. The MDM2 mRNA was predicted as a target of sd/mir-605, and sd/mir-605 overexpression was found to cause a reduction in MDM2 protein. MDM2 is a regulator of P53 tumor suppressor, and sd/mir-605 transfection was shown to increase P53 activity without increasing P53 protein levels, suggesting the sdRNA relieves MDM2-mediated repression of P53. In 2014 a study by Huang et al. found that single nucleotide polymorphisms (SNPs) in the sd/miR-605 precursor were significantly correlated with a shorter prostate cancer biochemical recurrence for patients [48].

Chen et al. found in 2017 that sd/miR-605 was significantly downregulated in a panel of melanoma cell lines and patient samples. Overexpression of sd/miR-605 markedly decreased cell proliferation, colony formation, and soft-agar growth in melanoma cell lines while reducing tumor volume in a murine model. INPP4B mRNA was confirmed as a sd/miR-605 target. The authors found that overexpression of sd/miR-605 decreases cellular INPP4B, leading them to conclude that sd/miR-605 suppresses melanoma growth through the inhibition of INPP4B [49].

In 2016 Alhasan et al. measured small noncoding RNA expression in prostate cancer patient serum samples and found that sd/miR-605 expression was significantly lower in low-risk prostate cancer and normal tissue compared to high-risk prostate cancer [50]. This publication established that sd/miR-605 is detectable in patient serum and is negatively correlated with the aggressive prostate cancer phenotype. A 2017 study from Zhou et al. found that sd/miR-605 is significantly downregulated in prostate cancer cell lines and patient samples [51]. EN2 mRNA was confirmed to be a sd/miR-605 target, and sd/miR-605 mimic decreased EN2 protein levels while antisense sd/miR-605 enhanced EN2 production. Prostate cancer proliferation and invasion were significantly reduced by sd/miR-605 overexpression, and this effect was rescued by transfection with EN2 cDNA. Cell-cycle analysis via flow cytometry revealed that sd/miR-605 overexpression resulted in a higher count of prostate cancer cells arrested at G_0_/G_1_. Taken together, the authors concluded that sd/miR-605 functions as a prostate cancer tumor-suppressing RNA by downregulating EN2.

### 4.14. Sd/miR-16-1

S/dmiR-16-1 is one of the miRNAs first implicated in human cancer [1]. In 2002, a paper by Calin et al. found that the sd/miR-16-1 gene was deleted in more than 65% of chronic lymphocytotic leukemia (CLL). Northern blot analysis of sd/miR-16-1 expression in CLL patient cancer and normal tissue revealed that miR-16-1 is downregulated in cancer. Sd/miR-16-1 has since been verified as a tumor suppressor in a multitude of cancers, several of which have been detailed in this 2009 review by Aqeilan et al. [84]. In 2017, the sdRNA was demonstrated to function as a tumor-suppressing RNA in gastric cancer by Wang et al. [21]. The authors found sd/miR-16-1 to be downregulated in gastric cancer patient samples. Additionally, overexpression decreased gastric cancer cell migration, invasion, and colony formation in vitro while reducing tumor volume in vivo. Target engagement with the 3′-UTR of TWIST1 was confirmed via the Renilla luciferase assay. Interestingly, a 2018 paper by a separate group, Feng et al., discovered that sd/miR-16-1 also targets TWIST1 in non-small cell lung cancer (NSCLC) [22]. Overexpression inhibited NSCLC proliferation, migration, and invasion in vitro. Together these publications add to the numerous examples of sd/miR-16-1 functioning as a tumor-suppressing RNA.

In 2019, Maximov et al. published an article in which they found that sd/miR-16-1 was downregulated in osteosarcoma (OS) patient samples [23]. Overexpression of sd/miR-16-1 was tumor suppressive in vitro resulting in decreased colony formation and invasion while enhancing apoptosis and chemosensitivity. Increasing sd/miR-16-1 expression in a murine model of OS resulted in a reduction of tumor volume and weight. Target prediction followed by Renilla luciferase engagement confirmation identified the 3′-UTR of FGFR2 as sd/miR-16-1’s target. Even more recently, Ye et al. conducted a similar study in 2020 where they found that sd/miR-16-1 targets PGK1 to suppress breast cancer malignancy [24]. In sum, sd/miR-16-1 is among the most well studied sdRNAs in cancer where its role as a tumor-suppressing RNA has been cemented.

### 4.15. Sd/miR-27b

In 2012, Ishteiwy et al. found that sd/miR-27b is downregulated in castration-resistant prostate tumors compared to primary prostate cancer and normal tissue [25]. Increasing sd/miR-27b expression in CRPC cell lines reduced invasion, metastasis, and colony formation. While there was no confirmation of target interactions, the authors found that modulating sd/miR-27b expression had a subsequent impact on Rac1 activity and E-cadherin expression. These findings indicate a tumor-suppressive role for sd/miR-27b in CRPC whose mechanism may involve the tumor suppressor gene E-cadherin.

Motivated by prior studies linking sd/miR-27b as a tumor-suppressing RNA, Wan et al. published a study in 2014 where they identified sd/miR-27b as downregulated in NSCLC patient tissues and cell lines [26]. Overexpression of the sdRNA reduced NSCLC proliferation, colony formation, migration, and invasion in vitro. The mRNA target was confirmed to be LIMK1, which is an oncogene that has been implicated as an enhancer of metastatic potential [85]. Regulation of LIMK1 expression by sd/miR-27b therefore suppresses the NSCLC malignant phenotype.

A 2021 publication by Li et al. measured sd/miR-27b expression in patient bladder cancer and adjacent normal tissue, finding that the sdRNA had lower expression in cancer. Overexpression of sd/miR-27b in bladder cancer cell lines caused decreases in cell proliferation, invasion, and migration while enhancing apoptosis. Target engagement with the 3′-UTR of the transcription factor EN2 was confirmed in vitro, and overexpression of sd/miR-27b was capable of reducing cellular EN2 protein as measured via western blot. Therefore, in bladder cancer the sdRNA sd/miR-27b functions as a tumor-suppressing RNA by reducing expression of the EN2 oncogene [27].

### 4.16. Sd/miR-31

The diverse role of miR-31 in cancer has been well studied and is the subject of two excellent reviews both in 2013 and more recently in 2018 [28,29]. Some examples from the aforementioned reviews include sd/miR-31 acting as a tumor-promoting RNA in CRC, HNSCC, and lung cancer while acting as a tumor-suppressing RNA in glioblastoma, melanoma, and prostate cancer [30,31,32,33,74,75].

### 4.17. Sd/let-7g

The let-7g miRNA belongs to the let-7 family of miRNAs, all of which have been well studied in the context of human cancer. Reviews published in 2010, 2012, and 2017 detail the let-7 family as well as let-7g specifically [86,87,88]. One example from this includes an early publication from 2008 by Kumar et al. where let-7g was found to suppress the NSCLC phenotype both in vitro and in vivo through the RAS pathway [34]. Many publications implicating sd/let-7g as a tumor-suppressing RNA in cancer have emerged since the aforementioned review, including a 2019 publication by Chang et al. in colorectal cancer and a 2019 paper by Biamonte et al. in ovarian cancer [35,36].

### 4.18. Sd/miR-28

Sd/miR-28 has been identified as a tumor-suppressing RNA in several cancers, including a 2014 publication by Schneider et al. focused on B-cell lymphoma (BCL) [37]. The sdRNA was downregulated in patient BCL samples (n = 25) compared to normal B cells (n = 4), and sd/miR-28 overexpression in BCL cell lines suppressed proliferation and colony formation while enhancing apoptosis. Target engagement was confirmed for sd/miR-28 with the 3′-UTRs of MAD2L1, BAG1, RAP1B, and RAB23 mRNA. Taken together, sd/miR-28 suppresses the BCL phenotype by regulating the expression of four oncogenes involved in cell cycle progression and apoptosis.

Two additional papers linking sd/miR-28-5p as a tumor-suppressing RNA were published recently in 2020. Fazio et al. found that sd/miR-28-5p binds to the 5′-UTR of SREBF2 to inhibit prostate cancer malignancy [38]. A separate publication by Ma et al. discovered that sd/miR-28-5p binds to the 3′-UTR of WSB2 mRNA to suppress breast cancer tumor migration [39]. The available literature indicates that in the majority of cases, sd/miR-28 gene products function as tumor-suppressing agents in a variety of cancers of diverse tissue origin.

### 4.19. pi-sno74, pi-sno75, pi-sno44, pi-sno78, and pi-sno81

Small RNA-seq (smRNAseq) of patient breast cancer tissue identified five differentially expressed sno-derived piRNAs (pi-snos) within the long noncoding RNA (lncRNA) GAS5 locus [76]. These pi-sno’s included pi-sno74, pi-sno75, pi-sno44, pi-sno78, and pi-sno81 (Figure 6). Each of these were found to be downregulated in breast cancer compared to adjacent normal tissue. Microarray and PCR revealed that pi-sno75 upregulates expression of tumor necrosis factor (TNF)-related apoptosis-inducing ligand (TRAIL), a proapoptotic protein. In silico target prediction identified a locus of 169 basepairs (bp) upstream of the TRAIL transcription start site that was highly complementary to pi-sno75 with predicted thermodynamically stable binding. To further investigate this mechanism of action, knockout (KO) and Co-IPs were performed to identify PIWIL1/4 binding partners during pi-sno75 overexpression. This revealed that pi-sno75/PIWIL can specifically interact with the methyltransferase complex protein WDR5, thereby increasing TRAIL expression. In MCF7 cells, pi-sno75 overexpression greatly increased TRAIL expression and, in combination with doxorubicin treatment, enhanced apoptosis. Treatment with pi-sno75 alone resulted in marked reductions of tumor volume in MCF7 and MDA-MB-231 mouse xenografts. In summary, the sno-derived piRNA pi-sno75 recruits epigenetic machinery to specifically upregulate TRAIL, and thus functions as a tumor-suppressive sdRNA.

### 4.20. pi-sno78 (Sd78-3′)

In 2011, a study by Martens-Uzunova et al. that focused on identifying a miRNA expression signature of prostate cancer also identified significantly differentially expressed sdRNAs [89]. While the majority of reads aligned to defined microRNAs, sdRNAs were found to be predominantly upregulated in the metastatic LN-PCa compared to local prostate cancer. The authors then focused on canonical miRNAs, leaving these enticing putative sdRNA drivers of metastatic LN-PCa uncharacterized until their follow-up publication in 2015. In their 2015 publication, Martens-Uzunova et al. specifically focused on investigating the role of sdRNAs in prostate cancer progression [77]. Consistent with their 2011 study, they identified an sdRNA from the 3′ end of snoRNA78 “sd78-3′” that was upregulated in the more aggressive LN-PCa patient samples. By examining expression at multiple stages of PCa, the authors concluded that globally overexpressed sdRNAs, including sd78-3′, are already present at early stages of cancer but exhibit striking overexpression concordantly with malignant transformation. While the authors did not investigate the mechanism of action of sd78-3′, this sdRNA is actually the pi-sno78 characterized in the above study by He et al. [76].

### 4.21. piR-017061

Revisiting the 2015 publication by Muller et al. which queried small RNA expression in PDAC patient samples and normal pancreas controls, a total of 123 ncRNAs were found to be significantly differentially expressed [70]. This approach considered 45 noncoding RNAs identified as significantly down-regulated in PDAC, of which there were fourteen sdRNAs and a single sno-derived piRNA, piR-017061. 

## 5. SdRNAomes

In a large-scale pan-cancer analysis of TCGA patient small RNA sequencing data and normal tissue controls, Chow et al. created a pan-cancer “sdRNAome” which was used to investigate the role of sdRNAs in the context of tumor immunity and clinical outcomes [52]. Small RNAseq data from a total of 10,262 tumor samples with 675 adjacent normal samples across 32 cancer types were aligned to full-length human snoRNAs to generate the pan-cancer “sdRNAome”. To determine whether a relationship exists between sdRNA expression and tumor immunity, the sdRNAome was queried alongside expression of PD-L1, CD8^+^ T cell infiltration, GZMA (serine protease granzyme A) expression, and other factors such as tumor vascularization (EndothelialScore) and overall survival (OS). Taking together correlation with PD-L1 expression, CD8^+^ T cell abundance, GZMA expression, and patient survival to calculate an ImmuneSurv score, the authors identified 267 sdRNAs that had a score of at least two (fulfilling at least two of the ImmuneSurv categories) in at least one cancer type. After including copy number variation as a category, 133 sdRNAs were found to be significant in all five tested categories. In sum, this far-reaching study provides a detailed examination of the human sdRNAome in patient tumor samples. Critically, sdRNA expression is linked to tumor immunity and patient survival in distinct cancer types, providing evidence that sdRNAs could function as useful biomarkers to assess cancer response to immunotherapies and to predict patient survival. Since a large number of sdRNAs were associated with cancers in this study, several representative sdRNAs highlighted in the paper have been compiled into Table 1 for the sake of clarity.

In addition to this, other publications reviewed above have reported comprehensive sdRNAomes comprised of differentially expressed sdRNAs in their respective cancers. In 2015 Martens-Uzunova et al. identified 657 unique sdRNAs from C/D-box snoRNAs and 244 unique sdRNAs from H/ACA-box snoRNAs in their analysis of patient prostate cancer samples. Of these, they identified at least 78 to be significantly differentially expressed in prostate cancer [77] (Table 2). Additionally, in 2017 our group identified 10 unique sdRNAs in MDA-MB-231 breast cancer cells that form complexes with AGO (Table 2). Further, five of these were significantly overexpressed in MDA-MB-237 compared to MCF7 breast cancer [72]. Most recently, in 2022, our lab similarly described 38 unique sdRNAs differentially expressed in prostate cancer patient samples [73] (Table 2).

## 6. Discussion/Conclusions

The past decade has clearly demonstrated the existence and functional relevance of small RNAs arising from full length snoRNAs. This new class of ncRNA has a similar form and function to that of miRNAs, and sdRNAs differ from traditional miRNAs primarily in terms of transcriptional origin. Additionally, alternative processing pathways to DICER/DROSHA have been proposed for sdRNA genesis, potentially distinguishing sdRNAs further from their miRNA cousins. In this review we have collected and detailed the growing catalogue of sdRNAs implicated in human cancer, including both tumor-promoting and tumor-suppressing sdRNAs. Like miRNAs, sdRNAs post-transcriptionally regulate gene expression in a myriad of ways that can result in highly variable effects on oncogenesis and malignant pathology. Despite differences in genetic origin and some possible bioprocessing divergence, the majority of sdRNAs appear to be functionally identical to canonical miRNAs. What then, is the purpose of pointing out sdRNAs? The answer to this question is two-fold.

First, and of extreme importance, is the fact that sdRNAs are routinely discarded from miRNA databases and consequently omitted from any miRNA-focused studies that follow. Even with mounting evidence to the contrary, sdRNAs are still considered by many to be random degradation products without relevance to gene regulation. As shown in this study, many such misannotated sdRNAs clearly contribute to a variety of cancers. Omission of sdRNAs unnecessarily hinders efforts to discover valuable new ncRNA biomarkers and therapeutic targets. We stress that, instead of discarding sdRNAs, they deserve inclusion in miRNA databases. By bringing the wealth of cancer sdRNA literature to the light, we hope that researchers who aim to search for ncRNAs regulating cancer will include sdRNAs in their search moving forward.

Second, we simply do not know precisely how distinct sdRNAs and miRNAs are from one another. There is ample evidence that sdRNAs are frequently processed independently of the DROSHA/DICER pathway [90]. Still, the preponderance of evidence indicates that they are extremely similar, especially regarding their function. Indeed, we have identified a 200 kb locus on human chromosome 14 that produces about 50 snoRNAs and 50 miRNAs (Appendix A), indicating that these ncRNAs are often co-regulated and functionally similar.

As evidenced by the sdRNAs outlined in this review, we suggest that including and emphasizing sdRNAs in exploratory ncRNA cancer regulation studies as well as closing the knowledge gaps regarding what makes sdRNAs distinct, including but not limited to their alternative biogenesis and mechanism of action pathways, will be of significant value in enhancing our understanding of cancer gene regulation. Notably, sdRNAs may be thought of as the lowest hanging fruit in cancer genomics research as, to date, the principal miRNA discovery algorithms have simply, erroneously, disregarded any and all small RNAs aligning to snoRNAs (example text: “Aligned sequences with the following annotations were eliminated (as potential microRNAs): tRNA, snoRNA, …” [91]). A more complete understanding and more extensive cataloguing of sdRNAs in cancer will likely result in novel biomarkers and treatment targets that now lay only just out of reach. By appreciating sdRNAs as functionally relevant ncRNA molecules in cancer, a great deal of readily attainable information regarding cancer gene regulation and the molecules responsible for pathological phenotypes may soon be realized.

## Figures and Tables

**Figure 1 biomedicines-10-01819-f001:**
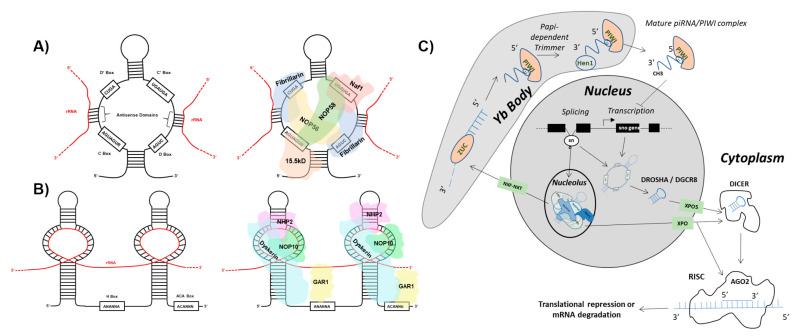
sdRNAs arise from full-length snoRNAs. (**A**) C/D box snoRNA structure and accessory proteins. (**Left**) The C/D box snoRNA consists of a 5′ C box (RUGAUGA) motif, a 3′ D box (CUGA) motif, and the C’ and D’ boxes located internally. Antisense domains identify target rRNAs (red) via complementarity. (**Right**) The C/D box snoRNP complex consists of NOP56, NOP58, 15.5 K, and fibrillarin proteins [4,5,6]. (**B**) H/ACA box snoRNA structure and accessory proteins. (**Left**) The H/ACA box snoRNA consists of a 3′ ACA box (ACANNN, N = any nt) and two hairpins that target rRNA (red) linked by an H box (ANANNA, N = any nt). (**Right**) The H/ACA box snoRNP complex consists of NHP2, NOP10, GAR1, and dyskerin proteins [7,8,9]. (**C**) sdRNA biogenesis and function. Biogenesis of Argonaute 2 (AGO2)-associating sno-derived RNAs (sdRNAs) Full length small-nucleolar RNAs (snoRNAs) are generated either as products of transcription or splicing [11,12,13,14]. snoRNAs produced by transcription can give rise to microRNA-like sdRNAs which are specifically excised from parent snoRNA transcripts by employment of the classical microRNA (miRNA) processing pathway. This occurs by processing of parent snoRNAs into smaller transcripts by the microprocessor complex which consists of Drosha Ribonuclease III (DROSHA) and DiGeorge syndrome critical region 8 (DGCR8). The intermediate snoRNA then undergoes cytoplasmic exportation via exportin 5 (XPO5). Following this, the smaller cytoplasmic snoRNA is processed by Dicer RNase III endonuclease (DICER) to generate the mature sdRNA which associates with AGO2, leading to the formation of the RNA-induced Silencing Complex (RISC). Similar to miRNAs, these sdRNAs function in post-transcriptional gene suppression by antisense binding to target mRNA transcripts within RISC [15]. That said, snoRNAs produced by splicing can also enter the classical miRNA processing pathway. Spliced snoRNAs, however, can bypass processing from DROSHA/DGCR8 and/or DICER as a result of trafficking to the nucleolus and subsequent processing by the fibrillarin complex followed by cytoplasmic export via a transporter belonging to the Exportin (XPO) family of proteins [10]. Biogenesis of PIWI-associated RNA (piRNA) like sdRNAs Spliced snoRNAs which arrive at the nucleolus for fibrillarin processing can be trafficked into Yb bodies via Nuclear RNA Export Factor (NXF1)/ Nuclear Transport Factor 2 Like Export Factor 1 (NXT1) where the 3′ end is cleaved by Zucchini (ZUC) and subsequently degraded. The remaining transcript is processed further within the Yb body by the papi-dependent trimmer. Following this, HEN1 double-stranded RNA binding protein binds at the 3′ end of the transcript where it adds a methyl group, generating a mature piRNA/PIWI complex which is exported to the cytoplasm. This piRNA/PIWI complex can then be shuttled back into the nucleus where it functions to inhibit transcription [16,17].

**Figure 2 biomedicines-10-01819-f002:**
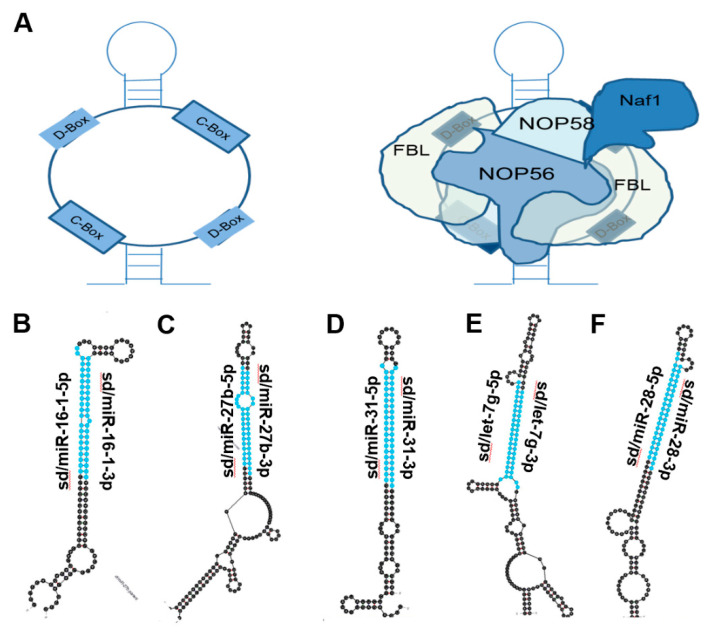
miRNAs derived from C/D Box snoRNA-like transcripts that bind fibrillarin. (**A**) Cartoon illustration of C/D Box snoRNA (**left**) and schematic of C/D Box snoRNA bound to fibrillarin complex, giving rise to snoRNP (**right**). Fibrillarin complex consists of NOP56, NOP58, NAF1 and Fibrillarin [18]. (**B**–**F**) Most thermodynamically stable secondary structures of C/D Box snoRNA-like transcripts described to bind fibrillarin complex [19] were obtained from mfold [20]. Mature miRNA sequences embedded in these transcripts are highlighted in blue. (**B**) miR16-1 (ENSG00000208006) is downregulated in chronic lymphocytic leukemia, gastric, NSCLC, Osteosarcoma, and breast cancer and is embedded within a C/D Box snoRNA-like transcript (GRCh38: chr 13:50048958-50049077:-1) known to bind fibrillarin [1,21,22,23,24]. (**C**) miR-27b (ENSG00000207864) is downregulated in prostate, lung, and bladder cancer and is located within a C/D Box snoRNA-like transcript (GRCh38: chr 9:95085436-95085592:1) known to bind fibrillarin [25,26,27]. (**D**) miR-31 (ENSG00000199177) is upregulated in colorectal, HNSCC, and lung cancer but is downregulated in glioblastoma, melanoma, and prostate cancer. miR31 is also located within a C/D Box snoRNA-like transcript (GRCh38: chr 9:21512102-21512221:-1) known to bind fibrillarin [28,29,30,31,32,33]. (**E**) miR-let7g (ENSG00000199150) has been shown to be downregulated in NSCLC, colorectal and ovarian cancers and is also embedded within a C/D Box snoRNA-like transcript (GRCh38: chr 3:52268239-52268408:-1) known to bind fibrillarin [34,35,36]. (**F**) miR-28 (ENSG00000207651) has been shown to be downregulated in B-cell lymphoma, prostate and breast cancer and is located within a C/D Box snoRNA-like transcript (GRCh38: chr 3:188688746-188688887:1) known to bind fibrillarin [37,38,39].

**Figure 3 biomedicines-10-01819-f003:**
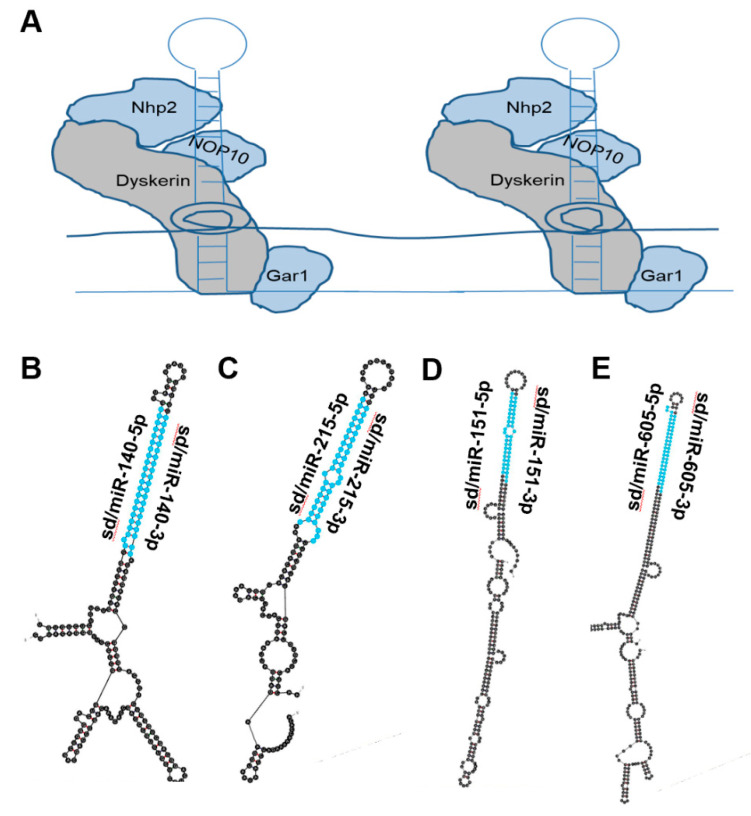
miRNAs derived from Box H/ACA snoRNA-like transcripts that bind dyskerin. (**A**) Schematic of box H/ACA snoRNA bound to dyskerin complex, giving rise to snoRNP. Dyskerin complex includes Dyskerin, NHP2, NOP10, and GAR1 [18]. (**B**–**E**) Most thermodynamically stable secondary structures of box H/ACA snoRNA-like transcripts known to bind dyskerin complex [40] were generated by mfold [20]. Mature miRNA sequences derived from these transcripts are highlighted in blue. (**B**) Box H/ACA snoRNA-like transcript (GRCh38: chr 16:69933072-69933264:1) was identified to bind dyskerin and encompasses miR-140 (ENSG00000208017) which functions as a tumor-suppressing RNA and is downregulated in prostate cancer [41]. (**C**) Box H/ACA snoRNA-like transcript (GRCh38: chr1:220117845-220118007:-1) was identified to bind dyskerin and surrounds miR-215 (ENSG00000207590) which is known to function as a tumor suppressor and is downregulated in ovarian, colorectal, prostate and lung cancer [42,43]. (**D**) Box H/ACA snoRNA-like transcript (GRCh38: chr8:140732552-140732807:-1) was determined to bind dyskerin and encapsulates miR-151 (ENSG00000254324) which is recognized as a tumor-suppressing RNA and is downregulated in prostate cancer [44,45,46]. (**E**) Box H/ACA snoRNA-like transcript (GRCh38: chr 10:51299393-51299708:1) was determined to bind dyskerin and surrounds miR-605 (ENSG00000207813) which is described to act as a tumor-suppressing RNA in melanoma, colorectal, breast, lung and prostate cancer [47,48,49,50,51].

**Figure 4 biomedicines-10-01819-f004:**
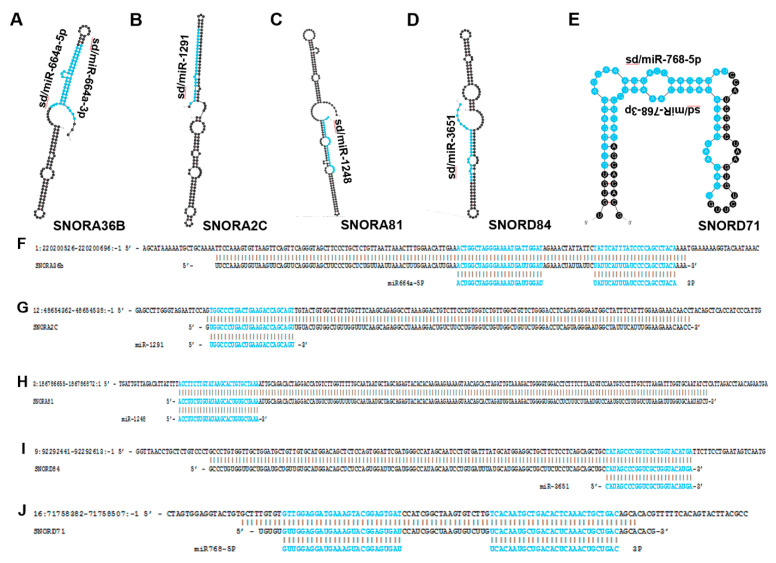
miRNAs that arise from snoRNA transcripts. The most thermodynamically stable secondary structures of snoRNA transcripts were generated by mfold [20]. Annotated miRNA sequences embedded in these transcripts are highlighted in blue (**A**–**E**). (**A**) miR-664a is embedded within the SNORA36B transcript and has been shown to function as a tumor-suppressing RNA in HCC and some types of cervical cancer while also functioning as a tumor-promoting RNA in cervical squamous cell carcinoma [56]. (**B**) miR-1291 is located within the SNORA2C transcript and is characterized as a tumor-suppressing RNA in pancreatic, prostate, breast and renal cell carcinoma [59,60,61,62,63]. (**C**) miR-1248 is derived from SNORA81 and has been determined to be upregulated in prostate cancer [64]. (**D**) miR-3651 is embedded within SNORA84 and is upregulated in colorectal cancer but downregulated in esophageal cancer [65,66]. (**E**) miR-768 resides within the SNORD71 transcript and has been shown to be downregulated in lung, breast, ovary, liver, parotid gland, thyroid gland cancers and melanoma [67,68,69]. (**F**–**J**) Mature miRNA sequences (highlighted in blue) were aligned to their corresponding snoRNA precursor transcripts and snoRNAs were aligned to their respective genomic regions. (**F**) Alignment between human genome (GRCh38: chr1:220200526-220200696:-1) (**top**), SNORA36B (ENSG00000222370) (**middle**), and miR664a (ENSG00000281696) (**bottom**). (**G**) Alignment between human genome (GRCh38: chr12:48654362-48654538:-1) (**top**), SNORA2C (ENSG00000221491) (**middle**) and miR-1291 (ENSG00000281842) (**bottom**). (**H**) Alignment between human genome (GRCh38: chr 3:186786655-186786872:1) (**top**), SNORA81 (ENSG00000221420) (**middle**), and miR-1248 (ENSG00000283958) (**bottom**). (**I**) Alignment between human genome (GRCh38: chr 9:92292441-92292613:-1) (**top**) SNORA84 (ENSG00000239183) (**middle**) and miR-3651 (ENSG00000281156) (**bottom**). (**J**) Alignment between human genome (GRCh38: chr 16:71758382-71758507:-1) (**top**), SNORD71 (ENSG00000223224) (**middle**) and miR-768 (ENSG00000223224) (**bottom**).

**Figure 5 biomedicines-10-01819-f005:**
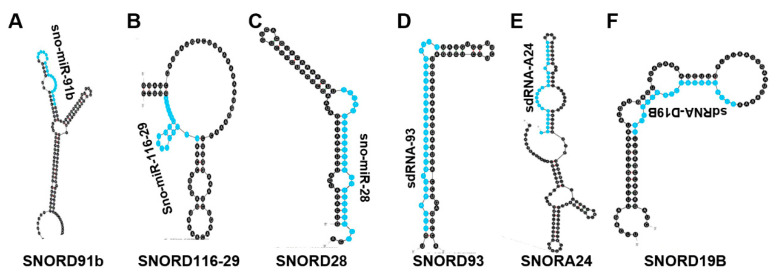
sdRNAs not Previously Annotated as miRNAs. (**A**–**F**) The most thermodynamically stable secondary structures of sdRNA producing snoRNA transcripts were generated by mfold [20]. Highlighted in blue are sdRNAs shown to function as bona fide miRNAs. (**A**) sdRNA produced by SNORD91B (ENSG00000275084) was determined to be downregulated in PDAC [70]. (**B**) sdRNA which arises from SNORD116-29 (ENSG00000207245) was determined to be downregulated in PDAC [70]. (**C**) sdRNA derived from SNORD28 (ENSG00000274544) was found to function as a tumor-promoting RNA and is upregulated in breast cancer []. (**D**) sdRNA excised from SNORD93 (ENSG00000221740) was found to function as a tumor-promoting RNA and is upregulated in breast cancer [72].(**E**) sdRNA produced from SNORA24 (ENSG00000275994) was functionally identified as a tumor-promoting RNA and is upregulated in prostate cancer [73]. (**F**) sdRNA derived from SNORD19B (ENSG00000238862) was determined to function as a tumor-promoting RNA and is upregulated in prostate cancer [73].

**Figure 6 biomedicines-10-01819-f006:**
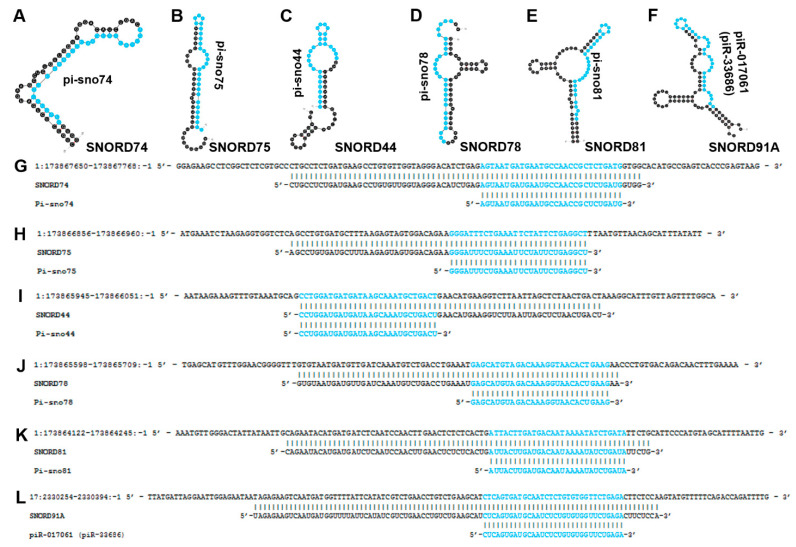
Sno-Derived Piwi-Interacting RNAs**.** (**A**–**F**) The most thermodynamically stable secondary structures of snoRNAs containing Piwi-interacting RNAs were obtained from mfold in which the piRNA sequence is highlighted in blue [20]. (**A**) pi-sno74 is embedded within SNORD74 (GRCh38: chr 1:173867674-173867745:-1) and is downregulated in breast cancer [76]. (**B**) Pi-sno75 is located within SNORD75 (GRCh38: chr 1:173866879-173866938:-1) and has been shown to be downregulated in breast cancer [76]. (**C**) pi-sno44 resides within SNORD44 (GRCh38: chr 1:173865968-173866028:-1) and is downregulated in breast cancer [76]. (**D**) pi-sno78 is embedded in SNORD78 (GRCh38: chr 1:173865622-173865686:-1) and is upregulated in prostate cancer metastasis but downregulated in breast cancer [76,77]. (**E**) pi-sno81 resides within SNORD81 (GRCh38: chr 1:173864146-173864222:-1) and is downregulated in breast cancer [76]. (**F**) piR-017061 (piR-33686) is embedded within SNORD91A (HBII-296A) (GRCh38: chr 17:2330279-2330370:-1) and was determined to be downregulated in PDAC [70]. (**G**–**L**) sno-derived piRNA sequences (highlighted in blue) were aligned to their respective snoRNA precursor sequences, and the snoRNA transcripts were aligned to their corresponding genomic loci. (**G**) Alignment between human genome (GRCh38: chr 1:173867650-173867768:-1) (**top**), SNORD74 (GRCh38: chr 1:173867674-173867745:-1) (**middle**) and pi-sno74 (**bottom**). (**H**) Alignment between human genome (GRCh38: chr1:173866856-173866960:-1) (**top**), SNORD75 (GRCh38: chr 1:173866879-173866938:-1) (**middle**), and pi-sno75 (**bottom**). (**I**) Alignment between human genome (GRCh38: chr1:173865945-173866051:-1) (**top**), SNORD44 (GRCh38: chr 1:173865968-173866028:-1) (**middle**), and pi-sno44 (**bottom**). (**J**) Alignment between human genome (GRCh38: chr1:173865598-173865709:-1) (**top**), SNORD78 (GRCh38: chr 1:173865622-173865686:-1) (**middle**) and pi-sno78 (**bottom**). (**K**) Alignment between human genome (GRCh38: chr1:173864122-173864245:-1) (**top**), SNORD81 (GRCh38: chr 1:173864146-173864222:-1) (**middle**) and pi-sno81 (**bottom**). (**L**) Alignment between human genome (GRCh38: chr17:2330254-2330394:-1) (**top**), SNORD91A (HBII-296A) (GRCh38: chr 17:2330279-2330370:-1) (**middle**) and piR-017061 (piR-33686) (**bottom**). (**A**–**E**,**G**–**K**) SNORD74, SNORD75, SNORD44, SNORD78, and SNORD81 arise from the GAS5 locus (ENSG00000234741) which is also annotated as a small nucleolar host gene (SNHG2).

**Table 1 biomedicines-10-01819-t001:** Summary of sdRNAs implicated in cancer.

sdRNA	Sequence (5′-sdRNA :: 3′-sdRNA) **	Parental snoRNA	Annotated as miR?	Cancer	Expression	Phenotypic Effect	Target	Reference
sdRNAs Misannotated as Traditional miRNAs								
**sd/miR-664a (ENSG00000281696)	5′-ACUGGCUAGGGAAAAUGAUUGGAU-3′ :: 5′-UAUUCAUUUAUCCCCAGCCUACA-3′	SNORA36B (ENSG00000222370)	Yes	Hepatocellular carcinoma	Downregulated in tumor	Tumor-suppressing	AKT2	[56]
				Cervical	Downregulated in tumor	Tumor-suppressing	c-Kit	[57]
				Cutaneous squamous cell carcinoma	Upregulated in Tumor	Tumor-promoting	IFR2	[58]
sd/miR-1291 (ENSG00000281842)	5′-UGGCCCUGACUGAAGACCAGCAGU-3′	SNORA2C (ENSG00000221491)	Yes	Pancreatic	Downregulated in Tumor	Tumor-suppressing	FOXA2	[59]
				Pancreatic	UNDETERMINED	Tumor-suppressing	FOXA2	[60]
				Renal Cell Carcinoma	Downregulated in Tumor	Tumor-suppressing	GLUT1	[61]
				Prostate	Downregulated in Tumor	Tumor-suppressing	MED1	[62]
				Breast	Downregulated in metastases	Tumor-suppressing	UNDETERMINED	[63]
sd/miR-1248 (ENSG00000283958)	5′-ACCUUCUUGUAUAAGCACUGUGCUAAA-3′	SNORA81 (ENSG00000221420)	Yes	Prostate	Upregulated aggressive tumor	UNDETERMINED	UNDETERMINED	[64]
sd/miR-3651 (ENSG00000281156)	5′-CAUAGCCCGGUCGCUGGUACAUGA-3′	SNORA84 (ENSG00000239183)	Yes	Colorectal	Upregulated in tumor	Tumor-promoting	TBX1	[65]
				Esophageal	Downregulated in tumor	UNDETERMINED	UNDETERMINED	[66]
**sd/miR-768 (ENSG00000223224)	5′-GUUGGAGGAUGAAAGUACGGAGUGAU-3′ :: 5′-UCACAAUGCUGACACUCAAACUGCUGAC-3′	SNORD71 (ENSG00000223224)	Yes	Breast	UNDETERMINED	UNDETERMINED	YB-1	[67]
				Gastric	Downregulated in tumor	UNDETERMINED	UNDETERMINED	[68]
				Lung, Breast, Ovary, Melanoma, Liver, Parotid Gland, Thyroid Gland, Large Cell	Downregulated in tumor	Tumor-suppressing, UNDETERMINED	KRAS	[69]
sdRNAs not Previously Annotated as miRNAs								
sd/hsa-sno-HBII-296B	NA	SNORD91B (ENSG00000275084)	No	Pancreatic ductal adenocarcinoma	Downregulated in Tumor	UNDETERMINED	UNDETERMINED	[70]
sd/hsa-sno-HBII-85-29	NA	SNORD116-29 (ENSG00000207245)	No	Pancreatic ductal adenocarcinoma	Downregulated in Tumor	UNDETERMINED	UNDETERMINED	[70]
sno-miR-28	5′-AAUAGCAUGUUAGAGUUCUGAUGG-3′	SNORD28 (ENSG00000274544)	No	Breast	Upregulated in Tumor	Tumor-promoting	TAF9B	[71]
sdRNA-93	5′-GCCAAGGAUGAGAACUCUAAUCUGAUUU-3′	SNORD93 (ENSG00000221740)	No	Breast	Upregulated in Tumor	Tumor-promoting	PIPOX	[72]
sdRNA-D19b	5′-AUUACAAGAUCCAACUCUGAU-3′	SNORD19b (ENSG00000238862)	No	Prostate	Upregulated in tumor	Tumor-promoting	CD44	[73]
sdRNA-A24	5′-CUCCAUGUAUCUUUGGGACCUGUCA-3′	SNORA24 (ENSG00000275994)	No	Prostate	Upregulated in tumor	Tumor-promoting	CDK12	[73]
miRNAs that Bind Dyskerin								
**sd/miR-140 (ENSG00000208017)	5′-CAGUGGUUUUACCCUAUGGUAG-3′ :: 5′-UACCACAGGGUAGAACCACGG-3′	Binds Dyskerin	Yes	Prostate	Downregulated in tumor	Tumor-suppressing	BIRC1	[41]
**sd/miR-151 (ENSG00000254324)	5′-UCGAGGAGCUCACAGUCUAGU-3′ :: 5′-CUAGACUGAAGCUCCUUGAGG-3′	Binds Dyskerin	Yes	Prostate	Downregulated in tumor	Tumor-suppressing	UNDETERMINED	[44]
**sd/miR-215 (ENSG00000207590)	5′-AUGACCUAUGAAUUGACAGAC-3′ :: 5′-UCUGUCAUUUCUUUAGGCCAAUA-3′	Binds Dyskerin	Yes	Ovary	Downregulated in Tumor	Tumor-suppressing	XIAP (not confirmed)	[42]
				Colorectal	Downregulated in Tumor	Tumor-suppressing	EREG, HOXB9	[43]
				Prostate	Downregulated in Tumor	Tumor-suppressing	PGK1 (not confirmed)	[45]
				Lung	Downregulated in Tumor	Tumor-suppressing	Leptin, SLC2A5	[46]
**sd/miR-605 (ENSG00000207813)	5′-UAAAUCCCAUGGUGCCUUCUCCU-3′ :: 5′-AGAAGGCACUAUGAGAUUUAGA-3′	Binds Dyskerin	Yes	Melanoma	UNDETERMINED	Tumor-suppressing	INPP4B	[49]
				Prostate	UNDETERMINED	Tumor-suppressing	EN2	[51]
				Colorectal, Breast Lung	UNDETERMINED	Tumor-suppressing, UNDETERMINED	Mdm2	[47]
				Prostate	Downregulated in tumor	UNDETERMINED	UNDETERMINED	[50]
				Prostate	UNDETERMINED	UNDETERMINED	UNDETERMINED	[48]
miRNAs that Bind Fibrillarin								
**sd/miR-16-1 (ENSG00000208006)	5′-UAGCAGCACGUAAAUAUUGGCG-3′ :: 5′-CCAGUAUUAACUGUGCUGCUGA-3′	Binds Fibrillarin	Yes	Chronic Lymphocytotic Leukemia	Downregulated in tumor	UNDETERMINED	Multiple (not confirmed)	[1]
				Gastric	Downregulated in tumor	Tumor-suppressing	TWIST1	[21]
				Non-small cell lung cancer	Downregulated in tumor	Tumor-suppressing	TWIST1	[22]
				Osteosarcoma	Downregulated in tumor	Tumor-suppressing	FGFR2	[23]
				Breast	Downregulated in tumor	Tumor-suppressing	PGK1	[24]
**sd/miR-27b (ENSG00000207864)	5′-AGAGCUUAGCUGAUUGGUGAAC-3′ :: 5′-UUCACAGUGGCUAAGUUCUGC-3′	Binds Fibrillarin	Yes	Prostate	Downregulated in tumor	Tumor-suppressing	UNDETERMINED	[25]
				Lung	Downregulated in tumor	Tumor-suppressing	LIMK1	[26]
				Bladder	Downregulated in tumor	Tumor-suppressing	EN2	[27]
**sd/miR-31 (ENSG00000199177)	5′-AGGCAAGAUGCUGGCAUAGCU-3′ :: 5′-UGCUAUGCCAACAUAUUGCCAU-3′	Binds Fibrillarin	Yes	Colorectal	Upregulated in tumor	Tumor-promoting	UNDETERMINED	[30]
				Head and neck squamous cell carcinoma	Upregulated in tumor	Tumor-promoting	FIH (not confirmed)	[31]
				Lung	Upregulated in tumor	Tumor-promoting	LATS2, PP2R2A	[74]
				Glioblastoma	Downregulated in tumor	Tumor-suppressing	RDX	[75]
				Melanoma	Downregulated in tumor	Tumor-suppressing	UNDETERMINED	[32]
				Prostate	Downregulated in tumor	Tumor-suppressing	UNDETERMINED	[33]
**sd/let-7g (ENSG00000199150)	5′-UGAGGUAGUAGUUUGUACAGUU-3′ :: 5′-UGAGGUAGUAGUUUGUACAGUU-3′	Binds Fibrillarin	Yes	Non-small cell lung cancer	Downregulated in tumor	Tumor-suppressing	KRAS (not confirmed)	[34]
				Colorectal	Downregulated in tumor	Tumor-suppressing	UNDETERMINED	[35]
				Ovary	Downregulated in tumor	Tumor-suppressing	UNDETERMINED	[36]
**sd/miR-28 (ENSG00000207651)	5′-AAGGAGCUCACAGUCUAUUGAG-3′ :: 5′-CACUAGAUUGUGAGCUCCUGGA-3′	Binds Fibrillarin	Yes	B-cell Lymphoma	Downregulated in tumor	Tumor-suppressing	MAD2L1, BAG1, RAP1B, RAB23	[37]
				Prostate	Downregulated in tumor	Tumor-suppressing	SREBF2	[38]
				Breast	Downregulated in tumor	Tumor-suppressing	WSB2	[39]
Sno-Derived Piwi-interacting RNAs								
pi-sno75	5′-GGGAUUUCUGAAAUUCUAUUCUGAGGCU-3′	SNORD75	No	Breast	Downregulated in Tumor	Tumor-suppressing	WDR5	[76]
pi-sno74	5′-AGUAAUGAUGAAUGCCAACCGCUCUGAUG-3′	SNORD74	No	Breast	Downregulated in Tumor	UNDETERMINED	UNDETERMINED	[76]
pi-sno44	5′-CCUGGAUGAUGAUAAGCAAAUGCUGACU-3′	SNORD44	No	Breast	Downregulated in Tumor	UNDETERMINED	UNDETERMINED	[76]
pi-sno78 (sd78-3′)	5′-GAGCAUGUAGACAAAGGUAACACUGAAG-3′	SNORD78	No	Breast	Downregulated in Tumor	UNDETERMINED	UNDETERMINED	[76]
				Prostate	Upregulated in metastases	UNDETERMINED	UNDETERMINED	[77]
pi-sno81	5′-AUUACUUGAUGACAAUAAAAUAUCUGAUA-3′	SNORD81	No	Breast	Downregulated in Tumor	UNDETERMINED	UNDETERMINED	[76]
piR-017061 (piR-33686)	5′-CUCAGUGAUGCAAUCUCUGUGUGGUUCUGAGA-3′	SNORD91A (ENSG00000212163)	No	Pancreatic ductal adenocarcinoma	Downregulated in Tumor	UNDETERMINED	UNDETERMINED	[70]

** For sdRNAs arising from two different loci on the same precursor, the 5′ sequence precedes “::” and the 3′ sequence follows. SdRNAs arising from just one locus are given as a single sequence.

**Table 2 biomedicines-10-01819-t002:** Cancer “sdRNAomes”.

Cancer	# of sdRNAs Identified	Experimental Validation	Reference
32 TCGA Cancer Types	>300	133 sdRNAs correlate with PD-L1 expression, CD8+ T cell abundance, GZMA expression, patient survival, and copy number variation	[52]
Prostate	78	sd78-3′ was found to be overexpressed in aggressive patient tumors	[77,89]
Breast	10	sdRNA-93 was confirmed to correlate with malignant invasion in vitro and cancer type in vivo	[72]
Prostate	38	sdRNA-D19b and sdRNA-A24 were confirmed to correlate with the malignant phenotype in vitro and cancer type in vivo	[73]

## Data Availability

Not applicable.

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
