# Peer review of "Small Nucleolar Derived RNAs as Regulators of Human Cancer"

_biomedicines, 2022, doi:10.3390/biomedicines10081819_

Round 1

Reviewer 1 Report

The review manuscript of Coley et al. aims to describe current knowledge in the field on the role of snoRNA-derived small RNAs (sdRNAs) in cancer. Although the paper provides a detailed overview of sdRNAs described in the literature and I cheer the enthusiasm of the authors and support their view of the importance of sdRNA, several major revisions and additions are needed before it can be considered for publication.

Introduction

Authors mention the importance of sdRNA as putative biomarkers, but the information provided in the review is focused on function. A sentence stating why understanding the role of sdRNAs is important for cancer biology needs to be added.

I strongly recommend to expand section 2 snoRNA function to snoRNA structure and function and provide clear description and illustration of the individual types of snoRNA C/D-box, HACA-box, including a concise description of the main structural and functional elements, i.e. C and D boxes H and ACA boxes and antisense box and proteins participating in the snoRNPs.  This would make the review more accessible for readers that are not familiar with RNA biology. In the current absence of this information figures 4 and 5 can be confusing and do not add to the value of the manuscript.

In section 3 Biogenesis, the concept of snoRNAs as evolutionary ancestors of miRNA is completely overlooked and does not appear in the paper while the tropic is well described in the literature and essential in the context of sdRNA biogenesis and function. The paper of Saraiya et al. 2008 PMID:19043559 is an important reference in this context as it also discusses the possibility to generate small RNAs without the involvement of Drosha and Exportin 5. Figure 1 needs to be redesigned as it currently appears that miRNA pathway is the major pathway to generate sdRNA. Although this may certainly be the case for some sdRNAs there are many that are generated via alternative paths, either by the piRNA pathway or by trimming of RNA parts exposed from the snoRNP. Existence of alternative yet unknown pathways is also not excluded. An important feature of sdRNA is their length. That is linked to the possible biogenesis mechanisms and should be indicated when discussing the different biogenesis pathways.  In this context, I disagree with the current wording of canonical and alternative sdRNA biogenesis and would rather refer to RNAi, PIWI, and snoRNA trimming as equally important for sdRNA biogenesis.

Section 4 sdRNA in cancer contains a lot of information which can be presented in more concise and structured way rather than as an inventory list. The table needs to be reformatted in landscape to allow for full sequence display on one row. Length in nt has to be included. It is not clear why some entries are presented as double stranded and others as single stranded. I can assume that has to do with the structural visualization listed in figures 2-6 but the connection is not described. All structural visualizations could be included in in a separate column of the table along with the corresponding sequence information. When listing sequences as double stranded it would be beneficial for the clarity if these are aligned rather in the :: format.

It is unclear why sequence alignments are included in fig 2 and 6. This can be provided as supplementary information.

Reviewer 2 Report

Review "Small Nucleolar Derived RNAs as Regulators of Human Cancer" by Alexander B. Coley and colleagues provide an interesting overview of sdRNA, their origin, biogenesis and nice catalogization. I like this manuscript - while I have doubts that such formal selection of small RNA based on transcriptional origin is not just descriptive as many of described RNA provide only minor effects in cancer development and the mechanisms are almost the same as for many "classic" miRNA. Nevertheless, the viewpoint is original and this review will be interesting for specialists in the field.

I would recommend to introduce more extensive comparison with miRNA for many features of sdRNA to emphasize the difference (if it really exist) between these two branches of small RNA 

Round 2

Reviewer 1 Report

In the revised version of the manuscript Coley et al. quickly attempted to improve the manuscript. Nevertheless, some of the most important flows of  the original version are still not addressed and require major revisions.

The authors insist, that according to their believes the miRNA processing pathway is the major sdRNA biogenesis pathway but do not cite references that provide the necessary mechanistic evidence supporting this statement. They provide the works of Kawaji (ref 2), Taft (ref 21), Scott (re f22), and Ono (ref 23), which are some of the first works in the field suggesting ancestral lineages and similar functions between sdRNA and miRNA. However, none of these papers suggests identical biogenesis pathways for the involvement of Drosha/Dicer in the biogenesis of sdRNA. The authors do not, present papers that provide mechanistic evidence. Furthermore, all of these works stress the existence of  separate populations of sdRNAs with characterised by their different size, indicative for the existence of alternative bioprocessing pathways as most nucleases produce fragments of rather specific length. The authors also seem to overlook the fact that although differences in length of  7 to 10 nucleotides  may appear less  important, they comprise for up to 30-40 % of the total size of sdRNAs and cannot be explained by promiscuity of the corresponding enzymatic complex. As exemplified by the large number of known miRNA, such promiscuity is also not characteristic for the miRNA processing pathway.

Next, the authors state that: “In this biogenesis pathway, full length snoRNAs are processed into shorter transcripts by the microprocessor complex composed of Drosha and DGCR-8, similar to the conversion of primary microRNAs 135 into precursor microRNAs20. Following this, the snoRNA fragment is exported from the 136 nucleus into the cytoplasm via Exportin-5 in a RAN-GTP dependent manner. Once in the 137 cytoplasm, the snoRNA fragment is processed further by the enzyme Dicer, yielding the 138 mature sdRNA which can then associate with Argonaute-2 to form the RNA Induced 139 Silencing Complex (RISC)15”. This statement is misleading. The provided reference by Gregory (ref 20) describes the microprocessor complex in the context of miRNA and does not mention snoRNAs at all, while the work of Ender (ref 15) specifically states the ACA45 is processed independently of Drosha/DGCR.

Taken together,  I have to conclude that presenting the miRNA processing pathway as the canonical pathway for generation of sdRNAs is inaccurate and misleading and has to be corrected according to my previous comments,  if the manuscript is to be published in a revised form.

The table is still in portrait view so I cannot assess if it has been improved in any way.

My suggestion to include sdRNA size in the table was discarded, despite the importance and information that it contains (in the context of the discussion above). However, the currently used “::” format is preserved which is not the most clear representation and can be misinterpreted.  A more clear representation would be to display sequences under each other, without the “::”;  in separate cells or with extra empty line between them. To save space references can be included in the same column with the target gene. To save space  and “declutter” he table, references can be included in the same column with the target gene.

“Tumor-promoter” should be replaced with “oncogene” .

Page 4, Line 125, 126. Add the reference to the work of Saraiya (ref 91) to the section on biogenesis “Consistent with this, there is evidence suggesting that H/ACA box and 125 C/D box snoRNA are the evolutionary ancestors of a subset of miRNA precursors 22,23.”

Although the authors disagree, I still do not see the practical use of presenting sequence alignments in the main figures in the manuscript. They are not annotated with sdRNA or snoRNA features and except for displaying the sequence (presented in the table) and the position (presented in the figures) of sdRNA do not highlight “necessary” nor supportive information. Furthermore, these are shown only for figures 2 and 6 but not for figures 3,4, and 5 which is inconsistent. Such alignments (for all figures) can be presented as a supplementary data.

Author Response

In this second round of feedback from Reviewer 1, we feel that most of the reviewer’s concerns were already addressed in the first round of revisions. This is detailed as follows:

     1)The Reviewer again raises the concern about Table 1 being presented in portrait format when we have clearly stated that the table will be resubmitted in landscape format by the editorial office.

     2)The Reviewer again expresses confusion about the sequences presented in Table 1, though we thoroughly explained this issue in round 1 and updated the table legend to further clarify that the sequences represent 5’ and 3’ sdRNAs, not a double stranded sequence.

     3)The Reviewer insists that “tumor–promoter” be replaced with the term “oncogene” in reference to sdRNAs. This is demonstrably incorrect, as oncogene specifically refers to genes that when mutated give rise to cancer (Oxford English Dictionary: https://www.oed.com/view/Entry/131331?redirectedFrom=oncogene#eid) (NIH: https://www.cancer.gov/publications/dictionaries/cancer-terms/def/oncogene). SdRNAs that promote the cancer phenotype are thus accurately described as “tumor-promoters”.

     4)Again the Reviewer requests that we remove sequence alignments from figures 2 and 6. We respectively disagreed, noting in the first round that “We believe that seeing the precursor transcript, parental snoRNA, and “miRNA” alignment provides necessary information that is supportive of, yet distinct from, what is communicated by the folded structures. For the reader, being able to see the alignment highlighted in blue that corresponds to the folded structure in the same figure provides valuable and readily-available context.”

     And finally,

     5)The Reviewer’s primary concern is related to our distinction between “canonical” and “alternative” sdRNA biogenesis. In the previous round we addressed his or her comment by stating:

     “We acknowledged the presence of alternative sdRNA biogenesis pathways in section 3 as well as in Figure 1, and we underscored in section 3 the fact that there are likely uncharacterized pathways yet to be discovered: “That said, there may be additional alternative pathways utilized in the biogenesis of sdRNAs which have not been realized”. We have revised Figure 1 to include more information about sdRNAs arising from snoRNAs, but we do believe that the miRNA-like pathway is the canonical pathway based on the available literature.”

     Despite this, the reviewer has determined that this requires major revisions. This Reviewer’s position is not grounded in the literature. sdRNAs were first discovered by search methods aimed at finding miRNAs, and were subsequently studied through the lens of miRNA biogenesis and function. In the first study of sdRNA biogenesis published by Ender /et al./ in 2008//(https://doi.org/10.1016/j.molcel.2008.10.017), they found sdRNAs “associated with human Ago1 and Ago2” whose processing “requires Dicer activity but is independent of Drosha/DGCR8”. Further, a study performed in mouse embryonic stem cells by Taft /et al./ in 2009 (https://doi.org/10.1261/rna.1528909) found that knocking down either DGCR8 (microprocessor) or DICER significantly reduced expression of 35/39 H/ACA sdRNAs (see Figure 3B in the paper). This highlights the significance of canonical miRNA processing machinery in sdRNA biogenesis and indicates the prevalence of sdRNAs that rely on this pathway. A 2011 study by Bramier /et al./ (https://doi.org/10.1093/nar/gkq776) identified 22 putative C/D box sdRNAs based on a search heuristic aimed at miRNA-like properties noting “These may serve as substrates for Dicer processing as described in ref. (27) and may enter the RNA silencing machinery as it is usually observed for (pre-)miRNAs”. Of these, 17 sdRNAs successfully silenced genes in a dual luciferase based reporter gene assay, supporting the search heuristic based on miRNA-like properties to identify functional sdRNAs. Many examples in the literature routinely show that functional sdRNAs associate with Ago via Ago pulldown including sno-miR-28 (Yu /et al,/ 2015, https://doi.org/10.1371/journal.pone.0129190), sdRNA-93 + 9 sdRNAs (Patterson/et al,/ 2017, https://doi.org/10.1038/s41523-017-0032-8), and sdRNA-A24 + -D19B (Coley /et al,/ 2022, https://doi.org/10.3390/cells11081302) to list a few.

     Our labeling of the miRNA-like biogenesis pathway as the canonical pathway is thus rooted in the literature. The presence of alternative biogenesis pathways that completely contradict the miRNA-like biogenesis pathway are noted in the review and included in our Figure 1. While important and worthy of inclusion in our review, the current literature does not suggest that the existence of alternative pathways usurps the status of miRNA-like biogenesis as the canonical sdRNA biogenesis pathway. Additionally, the striking similarity shared by sdRNAs and miRNAs has caused extensive study of evolutionary links between snoRNAs and miRNAs as reviewed by Scott /et al /in 2011 (https://doi.org/10.1016/j.biochi.2011.05.026). Supplementing the experimental evidence, this provides an evolutionary basis to support the notion that similarities between miRNAs and sdRNAs, including biogenesis pathways, are the standard against which other alternative pathways should be considered.

     That said, we have restructured portions of the introduction section 3.1 and 3.2 to provide enhanced clarity to this distinction between canonical and alternative biogenesis. As it now reads, all miRNA-like biogenesis (including DGCR8 or DICER independent processing that results in Ago-associated sdRNAs) is included under the category of “canonical processing” as these are slight deviations within the same broader miRNA-like pathway.